# Evaluating the Effects of Flooding Stress during Multiple Growth Stages in Soybean

Elizabeth Fletcher [1], Robert Patterson [2], Jeffery Dunne [2], Christopher Saski [3] and Benjamin Fallen [4,*]

[1] Department of Plant and Environmental Sciences, Virginia Polytechnic Institute and State University, Blacksburg, VA 24061, USA; fbodie1@vt.edu
[2] Department of Crop and Soil Sciences, North Carolina State University, Raleigh, NC 27695, USA; bob_patterson@ncsu.edu (R.P.); jcdunne@ncsu.edu (J.D.)
[3] Department of Plant and Environmental Sciences, Clemson University, Clemson, SC 29634, USA; saski@clemson.edu
[4] USDA-ARS Soybean Nitrogen Fixation Unit, Raleigh, NC 27607, USA
* Correspondence: ben.fallen@usda.gov

**Abstract:** Flooding is becoming an increasing concern for soybean (*Glycine max* [L.] Merr.) production worldwide due to the sensitivity of most cultivars grown today to flood stress. Flooding can stunt plant growth and limit yield, causing significant economic loss. One sustainable approach to improve performance under flood stress is to develop flood-tolerant soybean cultivars. This study was conducted to evaluate soybean genotypes for the response to flood stress at three critical growth stages of production—germination, early vegetative growth (V1 and V4), and early reproductive growth (R1). The results demonstrated that stress imposed by flooding significantly affected soybean yield for each growth stage studied. The average germination rate over the various treatments ranged from 95% to 46%. Despite the poor germination rates after the extended flood treatments, the flood-tolerant genotypes maintained a germination rate of >80% after 8 h of flooding. The germination rate of the susceptible genotypes was significantly lower, ranging 58–63%. Imposing flood stress at the V1 and V4 growth stage also resulted in significant differences between the tolerant and susceptible genotypes. Genotypes with the highest level of flood tolerance continually outperformed the susceptible genotypes with an average 30% decrease in foliar damage based on visual scoring and a 10% increase in biomass. The yield of the tolerant genotypes was also on average 25% higher compared to the susceptible genotypes. These results suggest that breeding for flood tolerance in soybean can increase resiliency during crucial growth stages and increase yield under flood conditions. In addition, the genotypes developed from this research can be used as breeding stock to further make improvements to flood tolerance in soybean.

**Keywords:** flood tolerance; germination; reproductive growth; vegetative growth; yield

## 1. Introduction

The United States (US) is consistently among the top three largest soybean-producing countries in the world [1]. Between 1961 and 2018, soybean production within the US increased by 570%—from 18.97 million tons to 123.66 million tons, respectively [2]. As such, soybeans are one of the main cash crops in the US, with approximately 83.1 million acres planted in 2020, and 1.6 million acres in North Carolina [3].

Flooding is just one of many natural occurrences with which producers must contend. Over the last 40 years, flooding alone has cost the US an estimated USD 161.6 billion in damages [4]. A study performed in 2019 concluded that between 2001 and 2016 more than 20 million hectares of soybeans in the US were lost due to damage caused by excess field moisture and flooding [5].

Excess water can lead to the reduced yield of many crops, as the standing water and water-logged soils deprive plants of the necessary light, oxygen, and carbon dioxide

required for growth [6]. Symptoms of flood stress in soybeans can range from reduction in nitrogen fixation within the root nodules, reduction in net photosynthesis, reduction in photosynthesis and chlorophyll synthesis-related genes, chlorosis and necrosis of the leaves, defoliation, stunting, and the most severe—plant death [7–12].

In areas prone to flooding, the yield loss from this environmental stressor can be just as detrimental as drought. The coastal plain region of North Carolina is the largest soybean-producing region of the state, and yield is often hampered by flooding. The soil of this region comprises a Portsmouth fine sandy loam, approximately 3.2% organic matter, and is situated on a high water table—making standing water at some point during the growing season a common event. In other regions in the nation, such as the Mississippi Delta, flooding during the early vegetative growth stages can cause a 25% reduction in yield [13–17].

The growth stage at which flooding occurs can have a significant impact on the plants' ability to respond and adapt to stress. Germination is a crucial stage, as it establishes the plant stand, and ultimately the yield potential. A study conducted by Wu et al. [17] evaluated germination under various flooding treatments. They concluded that flooding significantly affects germination rates and does not depend on genotype, flood tolerance, or yield potential. However, if flooding occurs after germination and the establishment of plant stands, the cultivar does play a more significant role in stress response and yield [18].

Scott et al. [18] demonstrated that plants exposed to temporary flooding during the early vegetative stages were more likely to produce a higher yield than plants exposed to flooding during the reproductive stages. Those exposed during the vegetative stages were able to recover much of the nitrogen (N) and potassium (K) lost three weeks post-flooding; however, recovery never reached non-flooded control levels. As a result, there was still a yield loss when compared to the control plots, but it was not as significant a loss as at the reproductive stages. The total yields for the V1, V4, and R1 flood trials were 88%, 83%, and 44% of the control yield, respectively [18].

Yield is also negatively impacted by the duration of flooding. Extended flooding at the V1 and V4 growth stages has been shown to significantly suppress root growth [9]. In the same study, the plant's growth stage, when exposed to extended flooding, also had a significant impact on root nodulation. Root nodulation was completely inhibited under flooding at the V1 stage and never recovered; however, those exposed at V4 could resume nodulation after the flooding had been removed [19]. Sallam and Scott [19] also observed that extended flooding at early vegetative stages caused stem cracking and reduced plant height. While the V4 trial only experienced 33.3% stem cracking compared to the 50% observed for V1, overall, the cracking at V4 was more severe than at V1.

Flooding can be unpredictable and occurs at different times throughout the growing season. Its impact on yield depends on the growth stage at which it occurs and for how long the fields remain under flood conditions. This study aimed to evaluate the performance of newly developed genotypes at various growth stages and durations of flooding in the North Carolina coastal plain region and to identify breeding lines that may be beneficial in developing soybean cultivars with improved flood tolerance. Previous studies have focused either on modeling [10], field screening at only one or two growth stages [9,14,18–20] or at germination [17]. However, to the knowledge of this article's authors, no study has evaluated more than two stages of development in the field. In addition, many of the genotypes identified as exhibiting flood tolerance are derived 12.5–50% by pedigree from exotic plant introductions and have not been previously reported.

## 2. Materials and Methods

### 2.1. Experiment I and II: Flooding Response at the V4 and R1 Growth Stages

In 2019, 2020, and 2021, two field studies were planted at the Tidewater Research Station (TRS) near Plymouth, NC (35°51′52.9″ N, 76°39′25.9″ W). This location consists of a Portsmouth fine sandy loam with approximately 3.2% organic matter, a flat landscape, and a high water table—all ideal for implementing flood treatments. Experiment I consisted

of 55 genotypes ranging in maturity from maturity groups (MGs) VI–VII. The genotype selections were based on previous flood stress observations made across multiple years and testing environments. In 2019 and 2020, the 55 selected genotypes were planted in a single row measuring 3 m in length with a row spacing of 7.62 cm.

Experiment II was grown in 2020 and 2021, with 15 genotypes evaluated, of which 5, 5, and 4, respectively, had previously been identified as tolerant, moderately tolerant, and susceptible to flood stress (Table 1). Of the 15 genotypes evaluated, 12 are derived from >12.5% wild soybean (*Glycine soja* Sieb. and Zucc.) by pedigree. In 2020, each genotype was grown in 3-row plots measuring 6.1 m in length with a row spacing of 7.62 cm. All data were collected from the center row. In 2021, each genotype was grown in 4-row plots measuring 3 m in length with a row-spacing of 5.08 cm. Each genotype was planted in a randomized complete block design (RCBD) with four replications for each experiment. Berms were constructed around each experiment to control the flooding.

**Table 1.** Visual ratings of flood stress injury to fifty-five genotypes evaluated in Plymouth, NC at the Tidewater Research Station in 2019 and 2020. Flooding was imposed at the V4 and R1 growth stage. Individual ratings were reported for each growth stage and combined across growth stages.

| | | 2019 | | 2020 | | | | |
| | | Flood Rating [†] | | | | | | |
| Genotype | Description | V4 [‡] | R1 [§] | V4 [‡] | R1 [§] | Mean V4 [‡] | Mean R1 [§] | Overall |
|---|---|---|---|---|---|---|---|---|
| NC-Dunphy | Flood Tolerate Check | 4.8 | 4.1 | 4.5 | 3.3 | 4.7 | 3.7 | 4.2 |
| Holladay | Flood Susceptible Check | 8.3 | 7.0 | 7.7 | 6.8 | 8.0 | 6.9 | 7.5 |
| HM06-204 | Flood Susceptible Check | 8.8 | 6.8 | 7.8 | 6.0 | 8.3 | 6.4 | 7.4 |
| PI 471938 | Moderate Fld. Tol. Check | 7.5 | 5.1 | 6.5 | 4.5 | 7.0 | 4.8 | 5.9 |
| Dillon | Moderate Fld. Tol. Check | 6.5 | 6.1 | 6.2 | 6.5 | 6.4 | 6.3 | 6.3 |
| N06-6 | Breeding Line | 7.0 | 5.4 | 6.5 | 5.5 | 6.8 | 5.5 | 6.1 |
| N8002 | Flood Tolerate Check | 4.8 | 4.3 | 5.1 | 4.0 | 5.0 | 4.2 | 4.6 |
| N93-110-6 | Breeding Line | 7.3 | 6.0 | 6.7 | 5.1 | 7.0 | 5.6 | 6.3 |
| N7002 | Flood Tolerate Check | 5.0 | 4.5 | 5.1 | 3.8 | 5.1 | 4.2 | 4.6 |
| N05-7364 | Breeding Line | 7.3 | 5.9 | 5.8 | 5.0 | 6.6 | 5.5 | 6.0 |
| N06-7164 | Breeding Line | 7.8 | 6.8 | 7.2 | 7.0 | 7.5 | 6.9 | 7.2 |
| N06-7274 | Breeding Line | 7.3 | 5.5 | 6.4 | 4.8 | 6.9 | 5.2 | 6.0 |
| NC-Roy | Yield Check | 7.8 | 5.8 | 6.2 | 5.8 | 7.0 | 5.8 | 6.4 |
| NTC94-5157 | Breeding Line | 7.5 | 5.5 | 6.6 | 5.0 | 7.1 | 5.3 | 6.2 |
| N05-7380 | Breeding Line | 5.3 | 4.6 | 5.2 | 4.3 | 5.3 | 4.5 | 4.9 |
| N06-7194 | Breeding Line | 7.5 | 6.0 | 6.1 | 5.5 | 6.8 | 5.8 | 6.3 |
| N07-15137 | Breeding Line | 7.0 | 6.8 | 6.4 | 5.6 | 6.7 | 6.2 | 6.5 |
| N07-15307 | Breeding Line | 8.5 | 6.9 | 7.0 | 7.5 | 7.8 | 7.2 | 7.5 |
| N09-13890 | Breeding Line | 6.8 | 5.9 | 5.8 | 6.0 | 6.3 | 6.0 | 6.1 |
| N10-7277 | Breeding Line | 7.3 | 5.4 | 6.2 | 5.0 | 6.8 | 5.2 | 6.0 |
| N10-7365 | Breeding Line | 8.0 | 6.0 | 6.6 | 5.0 | 7.3 | 5.5 | 6.4 |
| N10-7404 | Breeding Line | 7.3 | 6.0 | 5.9 | 5.5 | 6.6 | 5.8 | 6.2 |
| N10-7412 | Breeding Line | 6.8 | 5.3 | 5.7 | 5.1 | 6.3 | 5.2 | 5.7 |
| N10-7419 | Breeding Line | 7.3 | 5.9 | 5.8 | 5.3 | 6.6 | 5.6 | 6.1 |
| N11-10295 | Breeding Line | 4.8 | 3.5 | 4.7 | 3.8 | 4.8 | 3.7 | 4.2 |
| N06-7023 | Breeding Line | 7.5 | 6.3 | 6.4 | 5.4 | 7.0 | 5.9 | 6.4 |
| N01-11771 | Breeding Line | 6.5 | 6.0 | 5.1 | 7.3 | 5.8 | 6.7 | 6.2 |
| N01-11136 | Breeding Line | 7.5 | 6.1 | 6.2 | 5.3 | 6.9 | 5.7 | 6.3 |
| N11-7254 | Breeding Line | 6.8 | 5.2 | 5.6 | 4.5 | 6.2 | 4.9 | 5.5 |
| N11-7321 | Breeding Line | 6.8 | 5.6 | 6.0 | 5.5 | 6.4 | 5.6 | 6.0 |
| N11-7378 | Breeding Line | 7.3 | 6.1 | 6.0 | 5.5 | 6.7 | 5.8 | 6.2 |
| N11-7405 | Breeding Line | 7.0 | 6.0 | 6.3 | 6.8 | 6.7 | 6.4 | 6.5 |
| N11-7414 | Breeding Line | 8.5 | 7.7 | 7.8 | 8.0 | 8.2 | 7.9 | 8.0 |
| N11-7433 | Breeding Line | 5.8 | 5.1 | 5.0 | 4.6 | 5.4 | 4.9 | 5.1 |

**Table 1.** *Cont.*

| Genotype | Description | 2019 | | 2020 | | | | |
| | | Flood Rating [†] | | | | | | |
| | | V4 [‡] | R1 [§] | V4 [‡] | R1 [§] | Mean V4 [‡] | Mean R1 [§] | Overall |
|---|---|---|---|---|---|---|---|---|
| N11-7451 | Breeding Line | 6.5 | 6.5 | 6.0 | 5.5 | 6.3 | 6.0 | 6.1 |
| N11-7462 | Breeding Line | 6.8 | 6.1 | 5.9 | 5.1 | 6.4 | 5.6 | 6.0 |
| N11-7472 | Breeding Line | 6.5 | 5.8 | 5.3 | 6.0 | 5.9 | 5.9 | 5.9 |
| N11-7477 | Breeding Line | 6.8 | 6.5 | 6.3 | 5.5 | 6.6 | 6.0 | 6.3 |
| N11-7487 | Breeding Line | 6.0 | 5.3 | 5.3 | 4.4 | 5.7 | 4.9 | 5.3 |
| N11-7488 | Breeding Line | 6.3 | 6.0 | 5.3 | 6.3 | 5.8 | 6.2 | 6.0 |
| N11-7507 | Breeding Line | 6.8 | 5.4 | 6.2 | 5.0 | 6.5 | 5.2 | 5.9 |
| N11-7561 | Breeding Line | 6.3 | 6.8 | 6.1 | 5.6 | 6.2 | 6.2 | 6.2 |
| N11-7595 | Breeding Line | 7.8 | 7.2 | 7.0 | 6.5 | 7.4 | 6.9 | 7.1 |
| N11-7620 | Breeding Line | 8.0 | 7.2 | 7.1 | 6.1 | 7.6 | 6.7 | 7.1 |
| Woodruff | Yield Check | 6.3 | 5.8 | 5.1 | 5.3 | 5.7 | 5.6 | 5.6 |
| G00-3213 | Yield Check | 6.8 | 6.0 | 5.8 | 6.3 | 6.3 | 6.2 | 6.2 |
| N7003CN | Moderate Fld. Tol. Check | 6.5 | 5.2 | 5.7 | 5.2 | 6.1 | 5.2 | 5.7 |
| NC-Raleigh | Moderate Fld. Tol. Check | 6.3 | 5.0 | 4.8 | 5.3 | 5.6 | 5.2 | 5.4 |
| N09-12273 | Breeding Line | 6.3 | 6.4 | 5.7 | 5.0 | 6.0 | 5.7 | 5.9 |
| N11-9228 | Breeding Line | 7.3 | 6.5 | 6.8 | 7.0 | 7.1 | 6.8 | 6.9 |
| N11-12528 | Breeding Line | 7.0 | 5.7 | 5.7 | 5.3 | 6.4 | 5.5 | 5.9 |
| N14-8537 | Breeding Line | 5.8 | 6.0 | 5.3 | 5.3 | 5.6 | 5.7 | 5.6 |
| N11-352 | Breeding Line | 4.8 | 4.1 | 4.9 | 4.2 | 4.9 | 4.2 | 4.5 |
| N10-792 | Breeding Line | 5.0 | 3.3 | 5.0 | 3.5 | 5.0 | 3.4 | 4.2 |
| N16-9211 | Breeding Line | 6.5 | 5.6 | 5.4 | 5.4 | 6.0 | 5.5 | 5.7 |
| Mean | . | 6.8 | 5.8 | 6.0 | 5.4 | 6.4 | 5.6 | 6.0 |
| LSD$_{0.05}$ | . | 2.5 | 2.1 | 2.1 | 1.9 | 2.3 | 1.8 | 2.2 |

† Visual ratings on provided on a 0-to-9 scale: 0 = no damage, 1–2 = slight yellowing, 3–4 = minor yellowing, 5 = moderate yellowing and canopy defoliation, 6–7 = extensive yellowing and defoliation, 7–8 = severe chlorosis, and 9 = >95% severe chlorosis and plant death. ‡ Flooding was induced at the V4 vegetative growth stage, identified when the fourth trifoliate leaf unfolds. § Flooding was induced at the R1 reproductive growth stage, identified as when flowering starts.

The plots were each subjected to 4–6 cm of standing water for approximately 7 days. Visual ratings, on a scale of 0–9, were recorded 7 d and 14 d after the flood was released, a rating of 0 being no visual symptoms and 9 indicating that ≥95% of plants were dead. Ratings of 1, 3, 5, and 7 indicated no damage, slight yellowing of leaves, moderate yellowing and defoliation of the canopy, and extreme yellowing and defoliation, respectively. Ratings were recorded at the vegetative (V) 4 stage and reproductive (R) 1 stage. The plant was considered at the V4 stage when the 4th trifoliate leaves were fully developed. The R1 growth stage was defined as when flowering began at any node on the main stem.

### 2.2. Experiment III: Early Development Evaluation under Flooding Stress

In 2021, an additional study was conducted at the TRS near Plymouth, NC, in which eight blocks were planted. Berms were constructed around each block using a 3-point inverted disc plow mounted to a tractor. Each berm measured 0.75 m in height, and 1 m in width. Within each block, each genotype was planted within a randomized complete block design (RCBD) with four replications. A total of 100 seeds were planted for each genotype in a single row measuring 3 m in length and a row spacing of 5.08 cm. Four of the blocks were used to measure flood response at germination and the remaining four to measure response at the V1 growth stage. In the same field, two control tests were grown, with no flooding and no berm construction.

To evaluate response at germination, the berms were flooded to 4–6 cm of standing water, three days after planting. At germination, four flooding treatments were imposed: 8 h, 16 h, 24 h, and 36 h. Germination rates were recorded 14 days (d) after the release of flooding and were determined by counting the number of emerged seedlings from the 100 seeds that were planted.

To evaluate the effects of flood stress at the V1 growth stage, 15 genotypes were planted within the four remaining blocks with surrounding berms. The plants were considered to be at the V1 growth stage when leaves were fully developed at the unifoliate node. Upon reaching this stage, the berms were flooded to 4–6 cm above the base of the plant and the levels maintained for 3 d, 6 d, and 10 d. Genotypes were evaluated and plant height in cm recorded upon reaching the V5 growth stage. Plant height was defined as the distance from ground level to the apical meristem. Flood scores were also taken at the V4 and R1 growth stages using the same 0–9 visual rating scale as described previously. The dry biomass at the R1 stage was then calculated from 15 plants in each row.

### 2.3. Statistical Analysis

Flooding treatments and genotypes were considered fixed effects; all other effects were treated as random. Statistical analyses were performed with SAS version 9.4 (SAS Institute Inc., Cary, NC, USA). Analysis of variance (ANOVA) and least square means (LSMEANS) were performed using PROC MIXED. Fischer's protected least significant difference (LSD) was used to calculate significant differences among different treatments with a confidence level of $p \leq 0.05$. Pearson correlation coefficients were calculated on an entry means basis to assess the relationships among yield, maturity, seed size, and visual flood ratings for flooded and non-flooded treatments.

## 3. Results

### 3.1. Experiment I

Of the 55 genotypes evaluated for flood tolerance (Table 1), N11-10295 and N10-792 performed the best with a mean flood rating of 4.2 for both growth stages. N05-7380 (4.9), N11-7433 (5.1), and N11-352 (4.5) also had overall scores statically similar to the flood tolerate checks NC-Dunphy (4.2), N8002 (4.6) and N7002 (4.6). The flood-susceptible checks Holladay and HM06-204 had an overall flood score of 7.5 and 7.4, respectively. N11-7414 showed the most flood stress injury Genotypes N11-7595 (7.1), N11-7620 (7.1) and N07-15307 (7.5) also exhibited overall flood scores statistically similar to the flood susceptible checks. From the 55 genotypes tested, 15 were then selected for further evaluation in Experiment II and III (Table 2) and are shown in bold in Table 1.

**Table 2.** Descriptive characteristics of fifteen soybean genotypes evaluated for flood stress response in 2020 and 2021 at the Tidewater Research Station in Plymouth, NC.

| Genotype | Pedigree | % Exotic Pedigree | % Exotic PIs in Pedigree | Flood Trait |
|---|---|---|---|---|
| N05-7380 | N7002 × N98-7265 | 35.5% | 12.5% PI 416937, 25% PI 471938 | Tolerate |
| N07-15307 | N98-7265 × N98-7018 | 50% | 50% PI 471938 | Susceptible |
| N09-13890 | TCPR-83 × N01-11136 | 37.5% | 12.5% PI 416937, 25% PI 407948 | Mod Tolerate |
| N10-792 | N03-893 × G00-3213 | 12.5% | 12.5% PI 416937 | Tolerate |
| N11-10295 | N01-11298 × N04-9646 | 12.5% | 12.5% PI 416937 | Tolerate |
| N11-352 | N05-741 × N05-196 | . [†] | . [†] | Tolerate |
| N11-7414 | NC-Roy × PI 587696 | 50% | 50% PI 587696 | Susceptible |
| N11-7433 | NC-Roy × PI 587563B | 50% | 50% PI 587563B | Mod Tolerate |
| N11-7595 | NC-Roy × PI 587563B | 50% | 50% PI 587563B | Susceptible |
| N11-7620 | NC-Roy × PI 587563B | 50% | 50% PI 587563B | Susceptible |
| N14-8537 | NMS4-44-329 × N7103 | 25% | 25% PI 366122 | Tolerate |
| N8002 | N7002 × N98-7264 | 37.5% | 25% PI 471938, 12.5% PI 416936 | Tolerate |
| NC-Dunphy | MD99-6226 × N97-9677 | 12.5% | 12.5% PI 416937 | Mod Tolerate |
| NC-Raleigh | N85-492 × N88-480 | . [†] | . [†] | Mod Tolerate |

† A dot (.) indicates no exotic germplasm was used in the development of this line.

### 3.2. Experiment II

In the 2020 field trial, flood scores at the V4 growth stage of the 15 genotypes tested ranged from 4.5 (N10-792 and N11-10295) to 7.5 (N11-7595) (Table 3). Overall, each variety performed better under flood conditions at the R1 stage, with flood scores ranging from 2.6 (N11-10295) to 8.0 (N11-7620). The three exceptions were N11-7414, N11-7620, and NC-Raleigh, in which all showed increased flood symptoms during the R1 stage. Genotypes N10-792 (4.1), N11-10295 (3.6), and N11-352 (4.6) had significantly lower mean visual rating than flood tolerate check NC-Dunphy (5.8). The same three genotypes also had a significantly lower flood score at R1 than NC-Dunphy. At V4, N10-792 (4.5) and N11-10295 (4.5) exhibited a flood score significantly lower than NC-Dunphy. N05-7380 also performed well with a mean visual rating and R1 flood score statistically similar to N8002 (4.8) and NC-Dunphy. However, at V4, N05-7380 (4.7) had a flood score statistically less than NC-Dunphy (6.0). Regarding yield, N10-792 (4428 kg/ha), N11-352 (4912 kg/ha) and N8002 (4703 kg/ha) exhibited the three highest yields under non-flooded conditions (Table 3).

**Table 3.** LS means of 15 genotypes in flooded and non-flooded (control) treatments at Plymouth, NC in 2020. The flooding treatment consisted of 4–6 cm of water above ground level for ~7 days. Yield data were not collected in 2020 due to excess rainfall during harvest.

| Genotype | Flooded, NC 2020 | | | | Non-Flooded, NC 2020 | | |
|---|---|---|---|---|---|---|---|
| | V4 [†] | R1 [‡] | Maturity Date Oct 1 = 1 [§] | Mean Visual Rating [¶] | Maturity Date Oct 1 = 1 [†] | Yield (kg/ha) | Seed Size (g/100 Seed) |
| N05-7380 | 4.7 | 4.6 | 38 | 4.7 | 35 | 4233 | 12.7 |
| N07-15307 | 6.8 | 6.2 | 30 | 6.5 | 28 | 3810 | 16.7 |
| N09-13890 | 6.0 | 5.2 | 28 | 5.6 | 27 | 4031 | 16.3 |
| N10-792 | 4.5 | 3.6 | 35 | 4.1 | 33 | 4428 | 17.1 |
| N11-10295 | 4.5 | 2.6 | 31 | 3.6 | 33 | 3460 | 12.9 |
| N11-352 | 5.5 | 3.6 | 27 | 4.6 | 27 | 4912 | 12.5 |
| N11-7414 | 6.5 | 6.6 | 29 | 6.6 | 33 | 3857 | 12.9 |
| N11-7433 | 6.0 | 5.6 | 31 | 5.8 | 32 | 3796 | 13.5 |
| N11-7595 | 7.5 | 6.6 | 31 | 7.1 | 26 | 3642 | 17.3 |
| N11-7620 | 7.3 | 8.0 | 36 | 7.7 | 33 | 3964 | 15.6 |
| N14-8537 | 5.2 | 4.5 | 29 | 4.9 | 28 | 3957 | 15.8 |
| N8002 | 5.4 | 4.2 | 41 | 4.8 | 39 | 4703 | 15.3 |
| NC-Dunphy | 6.0 | 5.6 | 29 | 5.8 | 27 | 4381 | 17.0 |
| NC-Raleigh | 5.3 | 5.8 | 33 | 5.6 | 33 | 4273 | 13.7 |
| PI 471938 | 5.7 | 5.4 | 37 | 5.6 | 35 | 3702 | 15.9 |
| Mean | 5.8 | 5.2 | 32 | 5.5 | 31 | 4077 | 15.0 |
| LSD$_{0.05}$ | 0.8 | 1.3 | 4 | 1.1 | 9 | 405 | 1.3 |

† Flooding was induced at the V4 vegetative growth stage, identified when the fourth trifoliate leaf unfolds. ‡ Flooding was induced at the R1 reproductive growth stage, identified when flowering starts. § Maturity was determined as days after October 1, where October 1 = 1. ¶ Visual ratings on given on a 0-to-9 scale: 0 = no damage, 1–2 = slight yellowing, 3–4 = minor yellowing, 5 = moderate yellowing and canopy defoliation, 6–7 = extensive yellowing and defoliation, 7–8 = severe chlorosis, and 9 = >95% severe chlorosis and plant death.

In the 2021 field trial the 15 genotypes had a mean V4 flood score 0.8 lower than that of 2020 (Tables 3 and 4). The mean R1 flood score for 2020 was 0.3 lower than 2021. NC-Dunphy showed less stress injury in 2021 at the V4 stage with only a 3.9 rating compared to 6.0 in 2020. N10-792, which had one of the lowest V4 ratings in 2020, performed better in 2021 with a rating of 4.5 and 3.8, respectively. In 2021 on average each genotype performed more poorly at the R1 stage than the V4 with the mean rating at V4 being 5.0 and R1, 5.5. For the genotype N05-7380, the rating between the two years showed the least difference with 2020 V4 and R1, being 4.7 and 4.6 in comparison to 2021's rating of 4.3 and 4.6.

**Table 4.** LS means of 15 genotypes in flooded and non-flooded (control) treatments at Plymouth, NC in 2021. The flooding treatment consisted of 4–6 cm of water above ground level for ~7 days.

| Genotype | Flooded, NC 2021 | | | | | Non-Flooded, NC 2021 | | |
| --- | --- | --- | --- | --- | --- | --- | --- | --- |
| | V4 † | R1 ‡ | Maturity Oct 1 = 1 § | Seed Yield (kg/ha) | Seed Size (g/100 Seed) | Maturity Oct 1 = 1 † | Seed Yield (ka/ha) | Seed Size (g/100 Seed) |
| N05-7380 | 4.3 | 4.6 | 38 | 1928 | 11.7 | 39 | 4871 | 13.1 |
| N07-15307 | 6.1 | 6.4 | 32 | 1693 | 14.2 | 37 | 4166 | 15.5 |
| N09-13890 | 5.9 | 5.8 | 31 | 1707 | 11.8 | 36 | 4253 | 14.1 |
| N10-792 | 3.4 | 4.8 | 37 | 2479 | 13.8 | 40 | 4844 | 16.4 |
| N11-10295 | 4.1 | 5.1 | 32 | 1834 | 12.6 | 38 | 4011 | 12.2 |
| N11-352 | 3.8 | 5.1 | 31 | 1908 | 12.7 | 39 | 3722 | 13.8 |
| N11-7414 | 5.9 | 6.5 | 32 | 1639 | 10.7 | 36 | 3427 | 13.2 |
| N11-7433 | 4.7 | 5.0 | 35 | 2150 | 10.7 | 41 | 3615 | 13.9 |
| N11-7595 | 6.1 | 6.5 | 32 | 1458 | 14.6 | 36 | 3608 | 17.0 |
| N11-7620 | 5.9 | 6.5 | 37 | 1250 | 11.0 | 39 | 3628 | 15.6 |
| N14-8537 | 5.2 | 5.8 | 31 | 1814 | 12.8 | 36 | 4495 | 16.3 |
| N8002 | 5.0 | 5.4 | 42 | 1498 | 12.3 | 42 | 4912 | 14.0 |
| NC-Dunphy | 3.9 | 4.5 | 31 | 2076 | 13.1 | 37 | 4307 | 15.8 |
| NC-Raleigh | 5.7 | 5.5 | 35 | 1613 | 10.8 | 38 | 4025 | 13.0 |
| PI471938 | 4.9 | 5.5 | 35 | 1908 | 11.7 | 33 | 3702 | 12.9 |
| Mean | 5.0 | 5.5 | 34 | 1797 | 12.3 | 38 | 4106 | 14.5 |
| LSD$_{0.05}$ | 0.8 | 0.4 | 4 | 284 | 1.2 | 6 | 468 | 1.4 |

† Flooding was induced at the V4 vegetative growth stage, identified when the fourth trifoliate leaf unfolds.
‡ Flooding was induced at the R1 reproductive growth stage, identified when flowering starts. § Maturity was determined as days after October 1, where October 1 = 1.

Statistically, N10-792 had the greatest yield, 2479 kg/ha and, numerically, the lowest flood score (3.4) under flooded conditions in 2021 (Table 4). In addition, N10-792 had a seed yield (4844 kg/ha) that was statistically similar to the highest yielding genotype, N8002 (4912 kg/ha), under non-flooded field conditions. In comparison, N11-7620, had the lowest recorded yield of 1250 kg/ha under flooded conditions and the highest recorded flood score at R1. N11-7595 also exhibited a flood score of 6.5 at R1 and had the second lowest yield, numerically, under flooded conditions (1458 kg/ha). N11-7433 yielded 2150 kg/ha under flooded conditions, statistically similar to N05-7380 (1928 kg/ha), N11-352 (1908 kg/ha), and NC-Dunphy (2076 kg/ha). In contrast, N11-7433 (3615 kg/ha) yielded statistically less than N05-7380 (4871 kg/ha) and NC-Dunphy (4307 kg/ha).

Pearson correlation coefficients among five parameters, including yield, maturity, seed size, and visual flood ratings for flooded and non-flooded treatments, are reported in Table 5. Most notably, there was a strong negative correlation between the visual ratings taken at the flooded V4 growth stage and flooded yield ($r = -0.79$, $p < 0.01$) and the visual ratings at the flooded R1 growth stage and flooded yield ($r = -0.75$, $p < 0.01$). This validates the visual scores that were used to rate plots for flood tolerance because a higher score indicates that more flood damage was observed. A strong positive correlation ($r = 0.89$, $p < 0.01$) was also observed between visual ratings taken at the flooded V4 growth stage and flooded R1 growth stage, suggesting most genotypes exhibit flood tolerance when flooding occurs at multiple growth stages.

**Table 5.** Pearson correlation coefficients for 15 genotypes in flooded and non-flooded (control) treatments at Plymouth, NC in 2021 for yield, maturity, seed size, and visual flood scores.

| Treatment | Trait | Non-Flooded Yield [†] | Non-Flooded Maturity [‡] | Non-Flooded Seed Size [§] | Flooded Yield [†] | Flooded Maturity [‡] | Flooded Seed Size [§] | V4 [‖] | R1 [‖] |
|---|---|---|---|---|---|---|---|---|---|
| **Non-Flooded** | | | | | | | | | |
| | **Yield [†]** | 1.00 | | | | | | | |
| | **Maturity [‡]** | 0.38 [#] | 1.00 | | | | | | |
| | **Seed Size [§]** | 0.15 | −0.01 | 1.00 | | | | | |
| **Flooded** | | | | | | | | | |
| | **Yield [†]** | 0.33 [#] | 0.16 [¶] | 0.01 | 1.00 | | | | |
| | **Maturity [‡]** | 0.49 [#] | 0.57 [#] | −0.15 | −0.06 | 1.00 | | | |
| | **Seed Size [§]** | 0.30 [#] | −0.09 | 0.64 [#] | 0.18 [¶] | −0.25 [#] | 1.00 | | |
| | **V4 [‖]** | −0.40 [#] | −0.37 [#] | 0.12 | −0.79 [#] | −0.15 | −0.15 | 1.00 | |
| | **R1 [‖]** | −0.51 [#] | −0.39 [#] | 0.26 [¶] | −0.75 [#] | −0.22 [#] | 0.02 | 0.89 [#] | 1.00 |

† Yield was reported as kg/ha. ‡ Maturity was determined as days after October 1, where October 1 = 1. § Seed size was reported as g per 100 seeds. ‖ Flooding was induced independently at the V4 vegetative growth stage, identified as when the fourth trifoliate leaf unfolds and at the R1 reproductive growth stage, identified as when flowering starts. Visual ratings assigned on a 0-to-9 scale: 0 = no damage, 1–2 = slight yellowing, 3–4 = minor yellowing, 5 = moderate yellowing and canopy defoliation, 6–7 = extensive yellowing and defoliation, 7–8 = severe chlorosis, and 9 = >95% severe chlorosis and plant death. ¶ significant at 0.05 probability level. # significant at 0.01 probability level.

### 3.3. Experiment III

The germination rates of the 15 genotypes are recorded in Table 6. The control group exposed to 0 h flooding, had germination rates ranging from 92% (N05-7380) to 98% (N8002 and NC-Dunphy). The germination rates of N05-7380 after each additional 8 h of flood exposure decreased from 92%, 83%, 76%, 71%, and 59%, respectively. Genotype N05-780 had the highest germination rate after 32 h of flooding (59%) of the other genotypes studied.

**Table 6.** Comparing seed germination rate means under four different flooding durations and a control test without flood stress of 15 genotypes at Plymouth, NC in 2021.

| | Duration of Flooding | | | | |
|---|---|---|---|---|---|
| **Genotype** | **0 h** | **8 h** | **16 h** | **24 h** | **32 h** |
| N05-7380 | 92% | 83% | 76% | 71% | 59% |
| N07-15307 | 95% | 58% | 60% | 46% | 31% |
| N09-13890 | 94% | 72% | 69% | 61% | 47% |
| N10-792 | 97% | 82% | 75% | 69% | 57% |
| N11-10295 | 93% | 80% | 74% | 69% | 57% |
| N11-352 | 94% | 78% | 75% | 64% | 49% |
| N11-7414 | 97% | 62% | 60% | 49% | 34% |
| N11-7433 | 93% | 72% | 68% | 61% | 48% |
| N11-7595 | 95% | 63% | 60% | 49% | 34% |
| N11-7620 | 96% | 63% | 59% | 49% | 35% |
| N14-8537 | 94% | 71% | 66% | 57% | 45% |
| N8002 | 98% | 78% | 74% | 61% | 45% |
| NC-Dunphy | 98% | 80% | 74% | 68% | 57% |
| NC-Raleigh | 96% | 71% | 66% | 57% | 44% |
| PI 471938 | 92% | 71% | 68% | 60% | 47% |

NC-Dunphy, a tolerant check, had a germination rate of 98% under 0 h and 57% under flooding for 32 h—2% less than N05-7380. The genotypes N07-15307, N11-7414, N11-7595, and N11-7620, after the 32 h flood treatment, had the lowest recorded germination rates of 31%, 34%, 34%, and 35%, respectively.

The biomass for each genotype under the four flood treatments are recorded in Table 7. Within the control group, the tolerant check N05-7380 and N8002 were significantly higher recorded biomass compared to the other genotypes tested. N8002, had the largest recorded biomass of 32.2 g and was significantly similar to N05-7380—31.2 g. The biomass for each

of the 15 genotypes decreased following each extended treatment (0 d, 3 d, 6 d, and 10 d). N05-780 maintained the most biomass out of the genotypes tested, with a mean loss of 23%. The flood tolerant checks, N8002 and NC-Dunphy, had a mean biomass loss of 29% and 26%, respectfully. Genotype N07-15307 had the most significant recorded mean loss of biomass, 35%.

**Table 7.** Biomass (g) recorded for 15 genotypes evaluated under flood stress for 3 to 10 days and a control test without flooding (0 days) at Plymouth, NC in 2021.

| Genotype | Duration of Flooding | | | | |
|---|---|---|---|---|---|
| | **0 d** | **3 d** | **6 d** | **10 d** | **Mean Loss [†]** |
| N05-7380 | 31.2 | 25.9 | 23.8 | 22.1 | 23% |
| N07-15307 | 27.2 | 18.4 | 19.1 | 15.4 | 35% |
| N09-13890 | 26.3 | 20.2 | 19.5 | 17.3 | 28% |
| N10-792 | 28.2 | 23.1 | 21.2 | 19.5 | 25% |
| N11-10295 | 26.8 | 21.4 | 19.9 | 18.5 | 26% |
| N11-352 | 24.6 | 19.3 | 18.4 | 15.6 | 28% |
| N11-7414 | 21.5 | 15.6 | 15.0 | 12.7 | 33% |
| N11-7433 | 22.3 | 17.2 | 16.3 | 14.7 | 28% |
| N11-7595 | 21.8 | 15.9 | 15.4 | 12.8 | 33% |
| N11-7620 | 24.6 | 18.0 | 17.1 | 14.5 | 33% |
| N14-8537 | 28.7 | 21.7 | 20.4 | 17.9 | 30% |
| N8002 | 32.3 | 25.1 | 23.8 | 19.6 | 29% |
| NC-Dunphy | 26.9 | 21.6 | 20.0 | 18.3 | 26% |
| NC-Raleigh | 25.8 | 19.5 | 18.2 | 16.1 | 30% |
| PI 471938 | 25.6 | 19.5 | 18.6 | 16.7 | 29% |
| Mean | 26.3 | 20.2 | 19.1 | 16.8 | 29% |
| LSD$_{0.05}$ | 3.1 | 3.0 | 2.6 | 2.6 | . |

[†] Mean loss reported as the average loss across flooding durations (3 to 10 days) compared to the control (0 days).

## 4. Discussion

Previous research has shown that the growth stage soybeans are exposed to flooding does have a significant impact on the severity of damage to plant growth and development [18,21]. In addition, a model developed to project the response of soybean to future climate scenarios showed that intense rain events had a greater negative impact on yield than a 25% increase in rainfall distributed over 1–3 months [22]. This further emphasizes the need to improve flood tolerance in soybean.

In this study, the mean flooding score was numerically higher, indicating more severe damage, at R1 than at V4 in 2021. However, in 2020, the opposite was observed. Previous research has shown that soybeans are more sensitive to flooding at the R growth stages compared to the V growth stages [23]. While the level of tolerance at any particular growth stage is important, the authors of this manuscript conclude that the overall performance of a genotype across growth stages is the best method to select for tolerant genotypes. This is further supported by the strong positive Pearson correlation coefficient between the flooded V4 and R1 ratings reported in this study.

Since during a growing season it is unknown when flood damage could occur, it is desirable to select for genotypes with a broad tolerance to flooding across multiple growth stages. Flood tolerant QTLs have been previously identified in exotic PI lines [20], and from this study, genotypes with exotic PI pedigrees that exhibited flood tolerance at germination and the V4 and R1 growth stages were identified. The PIs used to develop the genotypes found to be tolerant in this study have not been previously evaluated for flood tolerance and will be investigated in the future to identify new QTL for flood tolerance in soybean. While PI 471938 was only described as moderately flood tolerant, one of the tolerant experimental lines, N05-7380, and a tolerant cultivar, N8002, are each 25% derived from PI 471938 by pedigree. Other experimental lines with a similar exotic pedigree may also show promise for increased flood tolerance.

No genotype had a germination rate ≥80% when exposed to flood treatments longer than 8 h. However, the genotypes N05-7380, N10-792, N11-10295, and NC-Dunphy did exhibit a germination rate >80% after 8 h of flooding, which is the minimum seed rate for certified soybean seed, as set by the Association of Official Seed Certifying Agencies (AOSCA). Maintaining a germination rate of at least 80% has been shown to produce an efficient plant stand for the maximum yield potential [24].

Genotype N05-7380 consistently performed well and had the largest percentage of germination (59%) under the most severe flood treatment (32 h). While well below the minimum requirement of 80%, this demonstrates some promise as a breeding line for improved flood tolerance at germination.

N05-7380 also had the largest recorded biomass of the experimental lines tested under the 0 d of flooding and continued to have the largest recorded biomass under each of the flood treatments, resulting in a mean biomass loss of only 23% after 10 days of flooding. This was statistically significant when compared to the biomass of all tested genotypes, with the exception of N10-792 and N8002 (Table 6). N05-7380 statistically outperformed NC-Dunphy—a tolerant check—by maintaining 8% more yield and 3% more biomass (Tables 4 and 6). N05-7380 also had the second largest recorded control yield of 4871 ka/ha. The control yield of N8002, a tolerate check, was not significantly greater than N05-7380, however, the yield of N05-7380 under flooding was significantly greater than N8002 under flooding (Table 4).

N10-792, a genotype that also performed well in the germination trial, had a mean biomass loss of only 26%. N10-792 had the highest yield under flooded conditions, low visual scores, and high yield under non-flooded conditions. The performance of N10-792 makes it an excellent line for breeding programs to use for increasing flood tolerance while maintaining high yields under non-flood conditions. The development of flood-tolerant cultivars has recently been reported as an unfeasible approach to improving flood tolerance in soybean [12]. Thus, the results of this study are very promising and greatly contribute to improving soybean performance under flood stress.

## 5. Conclusions

The response of N05-7380 to flooding was the most consistent across the three experiments conducted. Its low visual stress ratings, high germination rate after 8 h flooding, and its yield under flooding, were equal to—or greater than—several of the cultivars used as flood-tolerant checks. Other experimental lines that were identified as flood tolerant also demonstrated an increased performance under multiple flood treatments at various growth stages compared to the susceptible genotypes. As such, N05-7380, N10-792, and other tolerant genotypes show promise for use as breeding lines in the future development of flood-tolerant cultivars by both public and private breeders. Breeding for flood tolerance is complex and requires identifying diverse genotypes across a wide range of maturity groups. Soybean maturity is used to classify soybeans and indicates the growing region best suited for growing a particular maturity group. Classification of soybean maturity is based on the period of time from planting to maturity because soybean is a photoperiod-sensitive crop. The maturity and exotic pedigree of the genotypes identified in this study offer new germplasm not previously reported.

**Author Contributions:** E.F. and B.F. conceived the work, contributed to the concept, design, and data collection. B.F. contributed to the statistical analysis of the data. E.F. wrote the manuscript. R.P., J.D. and C.S. supervised, improved, revised, and reviewed the article. All authors have read and agreed to the published version of the manuscript.

**Funding:** We would like to thank the United Soybean Board (2220-172-0154) and North Carolina Soybean Producers Association (1109-2021-2271) for providing financial support. We also thank the support staff of the Soybean and Nitrogen Fixation Unit and the Tidewater Research Station for their assistance with field management. Mention of trade names or commercial products in this publication is solely for the purpose of providing information and does not imply recommendation or endorsement by the USDA. The USDA is an equal opportunity provider and employer.

**Data Availability Statement:** Raw data are available upon request to Ben Fallen. Data have not been archived in a repository.

**Conflicts of Interest:** The authors declare no conflict of interest.

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
