# Peer review of "Evaluating the Effects of Flooding Stress during Multiple Growth Stages in Soybean"

_agronomy, doi:10.3390/agronomy13051243_

Round 1
Reviewer 1 Report
This study (agronomy-2318276) evaluated soybean genotypes for their response to flood stress at three critical growth stages and found that flooding significantly affected soybean yield at each stage studied. The results suggest that flood-tolerant genotypes can improve resiliency during crucial growth stages and increase yield under flood conditions. Although this study is interesting, it requires significant improvements before it can be considered for publication. Specifically, the material and methods section needs better explanation, the results section should be more straightforward and concise, and the discussion should compare the findings with the current literature.
Minor comments:
Please check and formatting following Author Instruction;
-Considerer in last sentence abstract add a future prospectively;
-Keywords in alphabetic order;
L98. Glycine soja (Sieb….)
Table 1. Add SE and appropriate statistical; In addition, update legend.
Table 2. what is “.” Dot? Please, add in legend.
Pease, add a correct statistical between treatments (genotypes);
The discussion requires improvements. For instance, it would be beneficial to discuss your results in relation to the literature and highlight the advancements you have made compared to the current available methods. Additionally, it is essential to include up-to-date references to support your arguments. Furthermore, there is a concern that only three references were cited throughout the entire discussion topic, which is insufficient.
In the conclusion, it is necessary to describe the most relevant findings of your study. As a suggestion, you could also elaborate on how your work contributes to the advancement of science and society, and provide insights into future perspectives of your research.
Best regards,
Reviewer 2 Report
Please open the attached file

Round 2
Reviewer 1 Report
Dear Authors, Thank you for addressing my comments and incorporating my suggestions. Regarding, Glycine max (need italic; its scienfic name, check again all manuscript). However, the quality of the article has improved significantly. Thus, I recommend accepting the manuscript in its present form. But check again scientific name! Best regards,